# Graphene-integrated mesh electronics with converged multifunctionality for tracking multimodal excitation-contraction dynamics in cardiac microtissues

Hongyan Gao[1], Zhien Wang[2], Feiyu Yang [3], Xiaoyu Wang[1], Siqi Wang[1], Quan Zhang[1], Xiaomeng Liu [1], Yubing Sun [3,4,5], Jing Kong [2] & Jun Yao [1,4,5] ✉

Cardiac microtissues provide a promising platform for disease modeling and developmental studies, which require the close monitoring of the multimodal excitation-contraction dynamics. However, no existing assessing tool can track these multimodal dynamics across the live tissue. We develop a tissue-like mesh bioelectronic system to track these multimodal dynamics. The mesh system has tissue-level softness and cell-level dimensions to enable stable embedment in the tissue. It is integrated with an array of graphene sensors, which uniquely converges both bioelectrical and biomechanical sensing functionalities in one device. The system achieves stable tracking of the excitation-contraction dynamics across the tissue and throughout the developmental process, offering comprehensive assessments for tissue maturation, drug effects, and disease modeling. It holds the promise to provide more accurate quantification of the functional, developmental, and pathophysiological states in cardiac tissues, creating an instrumental tool for improving tissue engineering and studies.

Cardiac diseases are the leading cause of human morbidity and mortality[1]. Cardiotoxicity is also the main side effect preventing one-in-three drugs from being used clinically[2]. The lack of a close animal model makes in vitro cardiac tissues the important alternative[3]. Cardiac tissue built from human stem cell-derived cardiomyocytes (SC-CMs) is most frequently used[3,4]. They are also capable of retaining patient information for studying genetic diseases and personalized drug screening[3,5]. Clinical transplantation of these tissues may offer long-term hope for heart disease treatments[6]. Compared to planar tissue culture, three-dimensional (3D) cardiac microtissues (CMTs) are preferred tissue models because they can recapitulate cell phenotypes, microenvironment, and cell-cell interactions closer to the living organ[7].

The assessment of developmental state, drug effect, and disease mechanism often requires the close monitoring of physiological responses[3–5]. The electrical and mechanical activities are the most relevant parameters[8]. Importantly, as these activities are inherently connected through the excitation-contraction (EC) coupling[9], the simultaneous measurements of the correlated dynamics are highly desirable[8]. For example, a single-modal measurement of the electrical or mechanical response only is inherently inadequate to differentiate drug effects that work on the opposite part or both. In other cases, the

[1]Department of Electrical and Computer Engineering, University of Massachusetts, Amherst, MA 01003, USA. [2]Department of Electrical Engineering and Computer Science, Massachusetts Institute of Technology, Cambridge, MA 02139, USA. [3]Department of Mechanical and Industrial Engineering, University of Massachusetts, Amherst, MA 01003, USA. [4]Institute for Applied Life Sciences, University of Massachusetts, Amherst, MA 01003, USA. [5]Department of Biomedical Engineering, University of Massachusetts, Amherst, MA 01003, USA. ✉e-mail: juny@umass.edu

dysfunction of myocytes in chronic myocardial infarction is linked to an impaired EC coupling[10], and many arrhythmias are caused by a weakened EC coupling[8]. Thus, spatial mapping by tracking the electrical or mechanical response only is limited in identifying the weakened EC link. For developmental studies, tissue maturity or aging is often better characterized with correlated EC dynamics (e.g., evolutional details in the contractile phases)[11,12] than the typical information of electrical or mechanical amplitude only.

Compared to planar tissue culture, CMTs introduce additional challenges to the effective assessment. Specifically, traditional optical means to track electrical activity (e.g., by voltage-sensitive dye)[13] and mechanical motion (e.g., by micro post array)[14] are now inherently limited to the surface region due to the opacity in the tissue. Electronic sensors integrated on a planar or flexible substrate, which work well with planar tissue culture[15–20], now fall into the same scenario. The recent development of flexible mesh electronics provides the opportunity to embed addressable sensors in deep tissue[21–30]. In particular, the confined ribbon feature brings the mechanical properties in the mesh close to the tissue, which reduces the invasiveness and prolongs the interfacing stability[28,29]. However, existing platforms can only probe the single modality of either electrical or mechanical activity only[21–31]. This may be adequate for some (e.g., neural) tissues[28] but falls short of characterizing cardiac systems, in which the correlated mechanical and electrical responses are of paramount importance as discussed before[8].

The limitation is not without cause. Almost all existing electronic sensors are unifunctional and can probe the single property of either electrical or mechanical response only[21–31]. The conventional way of combining two complementary sensor types, which may work with the planar substrate, is now inherently limited for a confined ribbon-substrate integration. Among the limited number of demonstrations of simultaneous EC recordings in planar tissue culture, none of them can be translated to mesh integration, because at least one of the sensor technologies is either inaccessible[32] to deep tissue or too big in device size[33].

Fundamentally, the strategy of combining two sensor types inevitably leads to heterogeneity in integration and/or signal acquisition, which increases the overall footprint in interconnects and devices. This may be accommodated by a surface interfacing but is particularly unfavored by the deep-tissue interfacing that requires minimized substrate size. Moreover, spatial separation between the two sensors inevitably introduces inaccuracy in signal correlation, which is unfavored as the EC coupling originates from a very localized cell level.

Recently, we developed a 2-in-1 sensor concept of converging electrical and mechanical sensing in one device, by exploiting the *field* effect and *piezoresistive* effect in a bottom-up semiconducting silicon nanowire[34]. The nanowire was specifically assembled into 3D suspended structure on a planar substrate to attain geometric freedom for enabling multifunctionality[34,35]. Both electrical action potential and mechanical contraction were simultaneously detected in planar cardiac tissue culture[34]. However, the device integration in CMT has not yet been achieved, given that attaining scalable 3D nanowire assembly from the bottom up on a freestanding substrate can be nontrivial. Overall, no existing sensing platform has realized the tracking of the multimodal EC dynamics in live CMTs.

## Results
### Basic device design and characterizations
We developed a sensing platform to overcome the above limitations. The overall theme was to still exploit converging the multifunctional sensing in one nanoelectronic sensor, integrate the sensor array in an ultra-flexible mesh electronics system, and innervate the system in the CMT (Fig. 1a). To facilitate scalable integration, the sensor was made from a graphene transistor (Fig. 1b–i) designed to uniquely achieve the

converged sensing of the multimodal EC dynamics. Specifically, on the one hand, the electrical cellular activity (e.g., action potential) can be detected by the transistor through the *field* effect[36] (Fig. 1b–ii). On the other hand, the strain induced by the mechanical cellular motion can be transmitted by the ultra-flexible ribbon substrate and electrically detected by the transistor through the *piezoresistive* effect[37] (Fig. 1b–iii). The ultra-flexible ribbon substrate is also important for avoiding the transfer of global bending to enable strict local motion detection. Together, one device can detect both electrical and mechanical activities simultaneously. The spatiotemporal convergence of both signals in the same device can enable the precise tracking of their correlated dynamics. The convergence can also lead to an equivalent reduction in overall device size and addressing budget, which is desirable for mesh integration.

The mesh electronics system was composed of a serpentine ribbon network to enable stretchability and accommodate tissue contraction. The ribbon was made of multiple layers (Fig. 1c; Figs. S1, S2). The bottom SU-8 layer (~400 nm thick) was lithographically defined for support and insulation. Monolayer graphene synthesized by chemical vapor deposition[38] (Fig. S3) was transferred onto the SU-8 layer, patterned into array, and electrically addressed by an interconnect layer (e.g., Pd/Au/Pd, 15/40/5 nm) to form the sensor array. The interconnects were passivated by a top dielectric layer (e.g., 70 nm $Si_3N_4$) for insulation[34]. All the materials used were studied to be biocompatible and stable in physiological environment[28,29,39]. The mesh was fabricated on a sacrificial layer (e.g., 120 nm Ge) to be dissolved for release, except for the input/output (I/O) contacts for connecting to the recording system.

The fabricated mesh covered an area of $5.8 \times 3.8\,mm^2$ (Fig. 1d), with a ribbon width ~20 μm (e.g., smaller than cell) and a spacing between neighboring ribbons ~200 μm (e.g., larger than cell) (Fig. 1e). The total thickness was ~530 nm (Fig. S4). These features yielded a highly porous system with a softness comparable to tissue[23], which is important for establishing intimate tissue interfacing and effective signal transduction. The size of the transistor (e.g., $20 \times 20\,μm^2$) was comparable to cell size to enable localized detection of cellular action potential and mechanical contraction (Fig. 1f). Raman spectrum confirmed the presence of the graphene transistor (Fig. 1g), showing the typical[38] *G* peak at 1594 $cm^{-1}$ and a *2D* peak at 2650 $cm^{-1}$ against the peak-less SU-8 substrate (gray curve).

Electrical characterizations were performed in the integrated transistors to reveal the potential for recording electrical and mechanical cellular responses. First, the electrical sensitivity was obtained by measuring the water-gate response in cell culture medium. The devices exhibited typical ambipolar behavior with the maximal transconductance ~$2.2 \pm 0.4$ mS/V (Fig. 1h; Fig. S5), consistent with typical values obtained on rigid substrates[36]. This enhanced transconductance (e.g., compared to values in silicon nanowire transistors[34]) can improve detection. Note that the sensor detection resolution is still affected by the noise level in individual devices. Our characterizations revealed the typical noise level ~20–40 nS (Fig. S5), equivalent to an estimated detection resolution ~30–60 μV (e.g., with a signal-to-noise ratio >3) that was sufficient for detecting extracellular action potentials[15]. The transconductance maintained stable value during repeated cycles of mesh folding (e.g., up to 180°) and stretching (e.g., 30% biaxial strain covering cardiac contractile range) (Fig. S6). Second, the mechanical sensitivity was obtained by measuring the piezoresistive response, which showed a linear decrease in conductance with the increase of bending stain (Figs. 1i, S7). The gauge factor (i.e., the ratio of fractional resistance change to the fractional strain change) was calculated to be ~$82 \pm 24$, consistent with the typical range observed previously[37]. With the relative noise level ~$10^{-5}$ (Fig. S5), the obtained gauge factor suggested that devices could detect a strain level <$10^{-6}$, sufficient for capturing cellular motion that typically induces local strain >$10^{-5}$ [34]. These properties suggested that the

integrated graphene transistors were capable of detecting cellular electrical and mechanical activities.

## Tissue integration and recording

The integrated mesh system was released in a chamber and fixed by a Matrigel layer, with the I/O contacts connected to the recording system (Figs. 2a, S8). Human embryonic stem cell-derived cardiomyocytes (hESC-CMs) were seeded on the mesh. During the continuous development, the tissue sheet grew, folded, and engulfed the mesh by the increasing cell-cell interactions to form a mesh-innervated CMT

(Figs. 2b, S9). The CMT continued to grow and maintained the integrity and contractile function across the entire developmental time of one month (Fig. 2c; Supplementary movie 1), showing the initial evidence of a stable mesh-tissue interface.

Multi-channel recordings revealed synchronized and periodic signals charactering the beating (~0.6 Hz) in the CMT (Fig. 2d). The recording yield (*e.g.*, 9/14 ~ 64%) was reasonable by taking consideration of yields in device fabrication (e.g., ~90–95%), functional cardiomyocytes (e.g., 80–98%) during differentiation[40], and potential inner necrotic part in the CMT resulted from supply limit of nutrient and

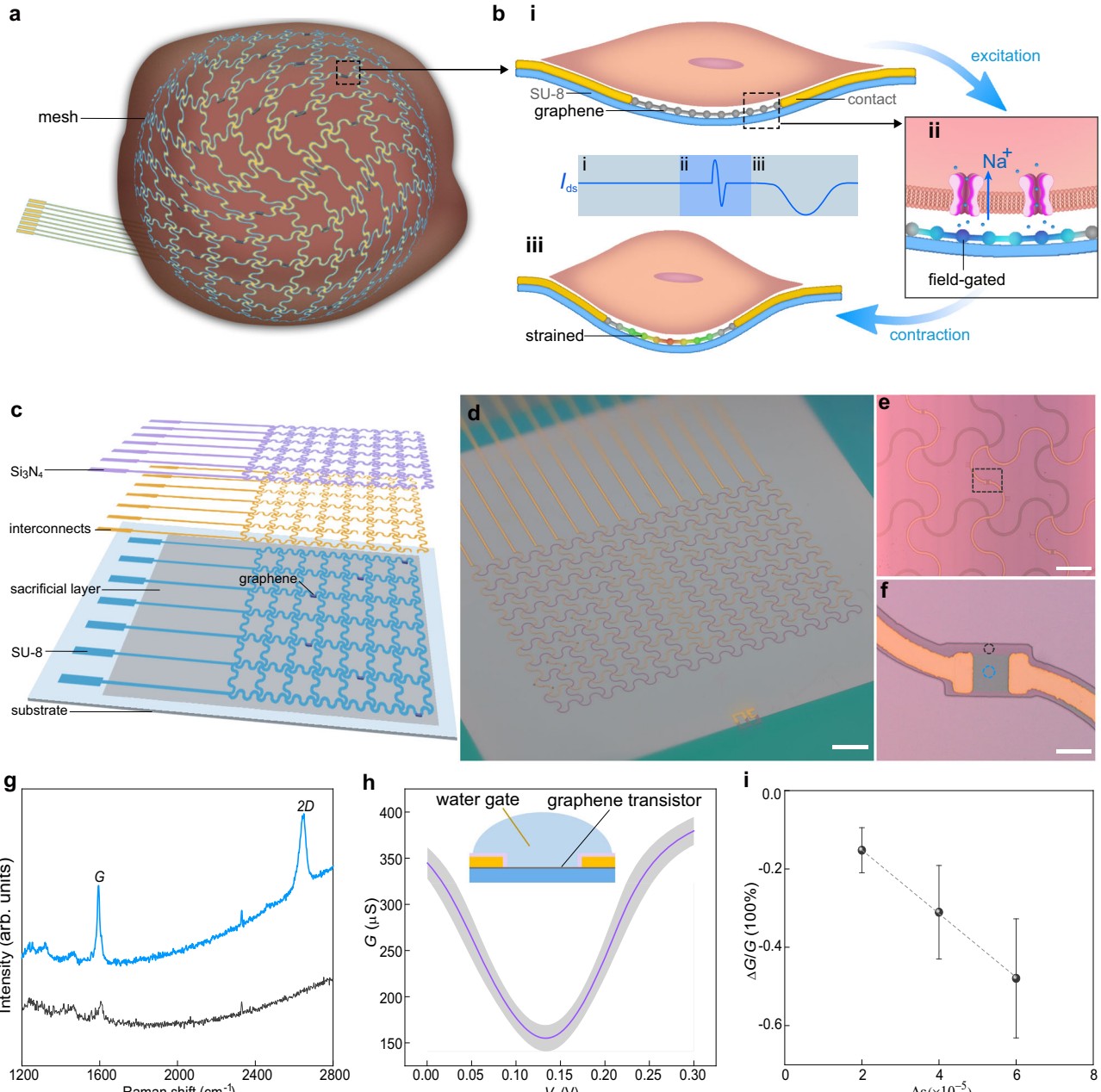

**Fig. 1 | Device concept and fabrication. a** Schematic of the mesh-innervated CMT. **b** Schematic of the (i) cell-device interface, with the graphene device detecting (ii) the action potential by the *field* effect and (iii) mechanical strain by the *piezoresistive* effect. The schematic $I_{ds}$ illustrates expected current change in each stage. **c** Schematic of the layered structure in the mesh. **d** Optical image of a fabricated mesh before the release from substrate. Scale bar, 400 μm. **e** Optical image of the ribbon feature, Scale bar, 200 μm. **f** Optical image of the graphene transistor in the dashed box in (**e**). Scale bar, 20 μm. **g** Raman spectra of the graphene (blue) and

SU-8 substrate (gray) at the locations indicated in (**f**). **h** Conductance (*G*)-water gate ($V_g$) relationship from 14 mesh-integrated graphene transistors mesh in Dulbecco's phosphate-buffered saline (DPBS) solution. The line and shadow represent the mean value and standard deviation, respectively. The inset shows the test setup schematic, in which a gold wire immersed in DPBS solution is used as the water gate. **i** Average relative conductance change (*ΔG/G*) in graphene transistors (*n* = 5 independent transistors) with respect to net strain. Data in (i) are presented as mean values ± SD. More details can be found in Fig. S7.

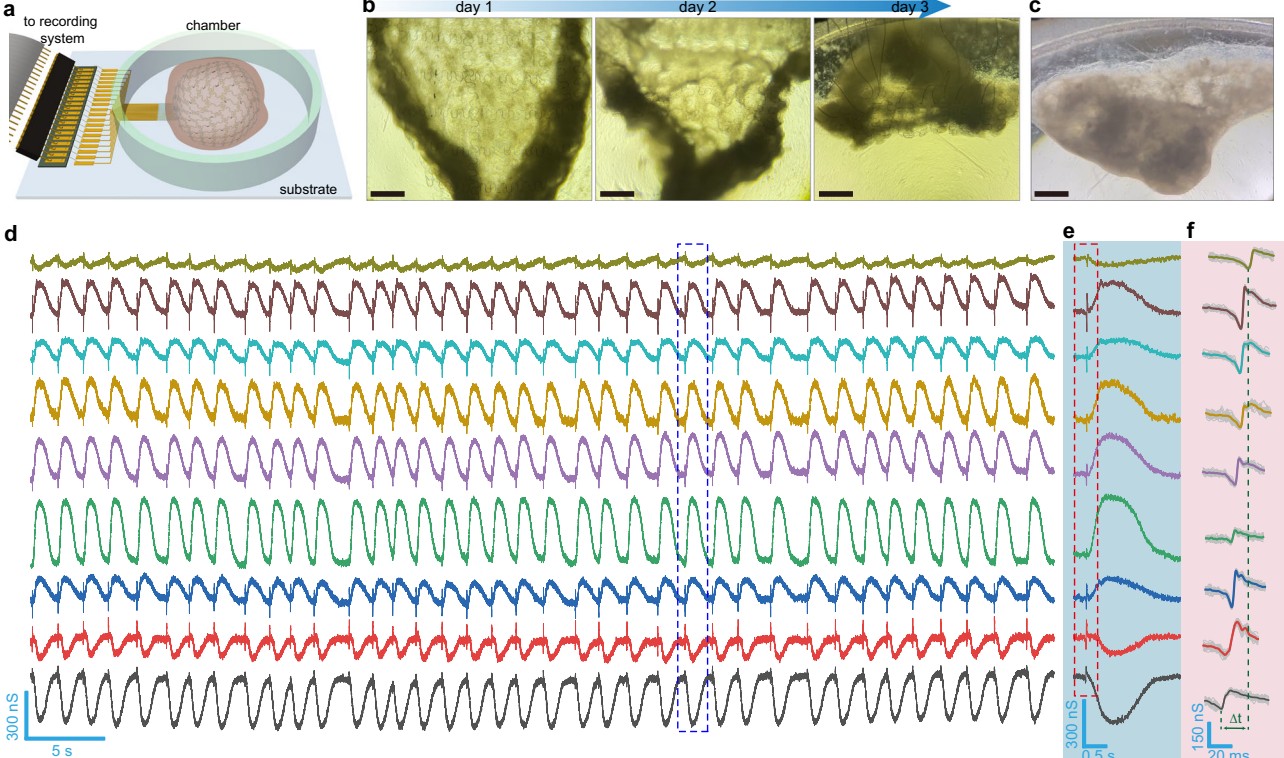

**Fig. 2 | Multi-channel recordings. a** Schematic of the setup for multi-channel recordings from the mesh-innervated CMT. A gold-wire electrode is immersed in the culture medium and grounded to serve as the global reference electrode. **b** Optical images of the formation process of the mesh-innervated CMT after 3 days of development (days 1, 2, 3 correspond to days 12, 13, 14 of cell differentiation). Scale bars, 500 μm. **c** Optical image of the mesh-innervated CMT after 30 days of development. Scale bar, 500 μm. **d** Recordings from 9 graphene sensors embedded in the CMT. **e** Zoom-in signals from the dashed box in (**d**). **f** Superimposed action-potential signals (dashed box in (**e**)) from the 9 channels. The two dash lines delineate the time latency (Δ*t*) between signals recorded from two devices.

oxygen by diffusion[41]. In each period, the broad peak was preceded by a sharp spike (dashed box, Fig. 2e). These sharp spikes showed a biphasic feature with the duration ~27 ± 6 ms (Fig. 2f) and calibrated amplitude of ~70–200 μV (Fig. S10), consistent with the properties of the extracellular action potential in hESC-CMs[24,27]. The noticeable variation in the shape features of the recorded action potentials can be attributed from the mixed cardiomyocyte subtypes differentiated from hESCs and cell states affected by diffusion gradient in oxygen and nutrient[41,42]. The time discrepancy between these action-potential signals (Fig. 2f) showed that the sensor array had a high temporal resolution to capture the propagating dynamics across different locations.

Additional investigations were carried out to correspond the broad peak to the mechanical response in the CMT. First, graphene transistors integrated on an unreleased mesh on the rigid substrate only detected the action-potential spikes but not the broad peak (Fig. S11), suggesting that the mechanical contraction was the source of the broad peak. Second, mesh containing only interconnects (e.g., without transistors) innervated in the CMT did not detect any signal (Fig. S12), confirming that the broad peaks were not mechanical artifacts from interconnects but active piezoresistive signals from the transistors. Third, shifting the gate reference ($V_g$) to move the graphene transistor from a *p*-type transport region to a *n*-type one reversed the signs in the broad peaks (Fig. S13). This is consistent with the general observation that the piezoresistive coefficient reverses the sign between *n*- and *p*-type conduction[43], which was further confirmed by the $V_g$-dependent strain response in the graphene transistors (Fig. S14). Finally, the broad peaks showed close evolution to that of the $Ca^{2+}$ dynamics governing the contractile dynamics in cardiomyocytes (Fig. S15)[44]. These results confirmed that the integrated graphene

transistors also captured the detailed mechanical responses in the CMT.

## Tracking maturation process

Improving maturation in CMT is key to recapitulating close organ function for precision studies[27,45]. An effective assessing means is indispensable to tissue engineering and studies for accelerating maturation[8]. Mature cardiomyocytes are generally characterized by concerted electrical and mechanical activities (e.g., fast depolarization, strong contraction, and close EC coupling)[7]. Previous unifunctional sensors only tracked one of these physiological components[21–29,31], falling short of a comprehensive assessment. The multifunctional graphene sensor could track all these components and provide an enriched set of parameters (Fig. 3a). Therefore, we intended to investigate if the integrated mesh system can provide a much more detailed tracking of the CMT development for the comprehensive assessment of maturation.

Continuous recordings across the entire maturing stage (e.g., > 1 month)[27,46] in the CMT were carried out. The transistor sensors maintained stable performance throughout the recording period (Fig. S16), forming the basis for capturing state evolution in the tissue. The key parameters charactering the electrical, mechanical activities, and their correlation (Fig. 3a) were extracted for the analysis of the CMT state. The recordings started from the 16th day of cell differentiation (corresponding to the 5th day of cell seeding on the mesh, Fig. S17). Representative recording showed a gradual increase in both the beating frequency *f* (e.g., from 0.5 to 1.4 Hz) and contractile amplitude $A_C$ (e.g., ~500% increase) during the process (Fig. 3b). An overlay of the mechanical signals also revealed a noticeable decrease in the contractile duration $t_2$, mainly contributed from the relaxation phase $t_5$

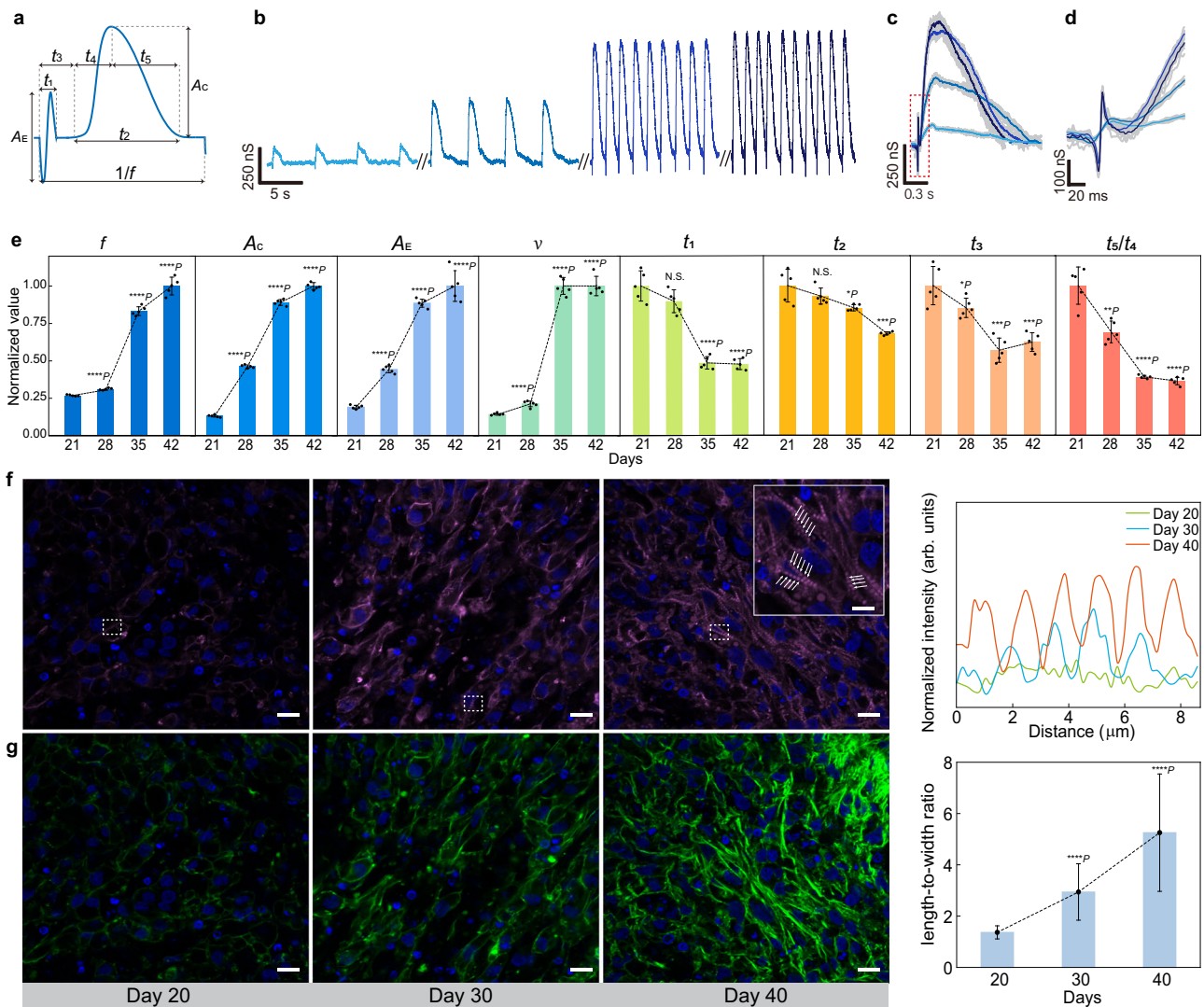

**Fig. 3 | Developmental characterization of CMT. a** Schematic of parameter sets in the recorded EC signal. **b** Representative recordings from the same device at days 21, 28, 35, and 42 of differentiation (corresponding to days 10, 17, 24, and 31 of seeding on the mesh). **c** Superimposed signals at these different days. The highlighted color lines represent the mean waveform. **d** Zoom-in signals from the dashed box in (**c**). **e** Evolution of the average ($n = 5$ independent recordings) parameter sets defined in (**a**) at days 21, 28, 35, and 42 of differentiation. All values were normalized to day-21 values. The conduction velocity ($v$) was revealed by measuring the signal delay between two sites (Fig. S18). Data in (**e**) are presented as mean values ± SD. *$P < 0.05$; **$P < 0.01$; ***$P < 0.001$, ****$P < 0.0001$, N.S. not significant, using one-way ANOVA with the day 21 group as control. **f** (Left) Fluorescence images of sliced CMT (30 μm thick) at days 20, 30, and 40 of differentiation. Blue and purple colors correspone to 4′,6-diamidino-2-phenylindole (DAPI) and cardiac Troponin T (cTnT), respectively. The 40-day tissue (inset, scale bar, 5 μm) shows improved sarcomere alignment (indicated by the white arrow arrays) with an average length of ~1.5 μm. Scable bars, 10 μm. (Right) Normalized fluorescence intensity from the dash boxes in left images. 5 samples were performed independently with similar results. **g** (Left) Fluorescence images of F-actin (green) in corresponding images shown in (**f**). Scable bars, 10 μm. (Right) Average ($n = 20$ cells examined over 5 independent samples) length-to-width ratios in cells at days 20, 30, and 40 of differentiation. Data in (**g**) are presented as mean values ± SD. ****$P < 0.0001$, using one-way ANOVA with the day 20 group as control.

(Fig. 3c). A zoom-in overlay of the action potentials revealed an increase in the amplitude $A_E$ but a decrease in the duration $t_1$ (Fig. 3d).

Statistical data from other devices featured the consistent trends of increasing beating frequency ($f$), contractile amplitude ($A_C$), action-potential amplitude ($A_E$), but decreasing action-potential duration ($t_1$) and contractile duration ($t_2$) mainly contributed from the decreasing relaxation phase ($t_5$) (Fig. 3e). These trends were consistent with the correlated cellular mechanisms in a maturing cardiomyocyte. Specifically, maturing cells are featured with an increasing rate of activation in the Na+ channels (e.g., upstroke velocity)[47], which was reflected from the shortening of the action-potential duration $t_1$. Maturing cells also recruit more Na+ channels[45] to yield an enhancing action-potential amplitude $A_E$ as observed. The increasing contractile amplitude $A_C$ suggested the improving contractile force contributed from the strengthening myofibrils, which was confirmed by fluorescence

imaging showing increasing intensity of cardiac troponin T (cTnT)[48] and improving sarcomere alignment (Fig. 3f, g). The multi-channel recordings also captured the temporal signal delay between different sites (Fig. S18), revealing an increasing conduction velocity ($v$) consistent with the enhancing electrical signaling in maturing tissue.

The simultaneous measurements of the action-potential and mechanical signals provided a unique window to peek into the Ca2+-handling dynamics underlying the EC coupling. A faster upstroke velocity in the action potential leads to the faster triggering of the L-type Ca2+ channel for extracellular Ca2+ recruitment[44], which further induces the Ca2+-induced Ca2+ release (CICR) in the sarcoplasmic reticulum (SR) for contraction. This improving EC coupling was uniquely captured by the integrated sensors, which showed a decreasing EC temporal delay ($t_3$). The convergence of both electrical and mechanical sensing at the same location ensured the precision in the EC

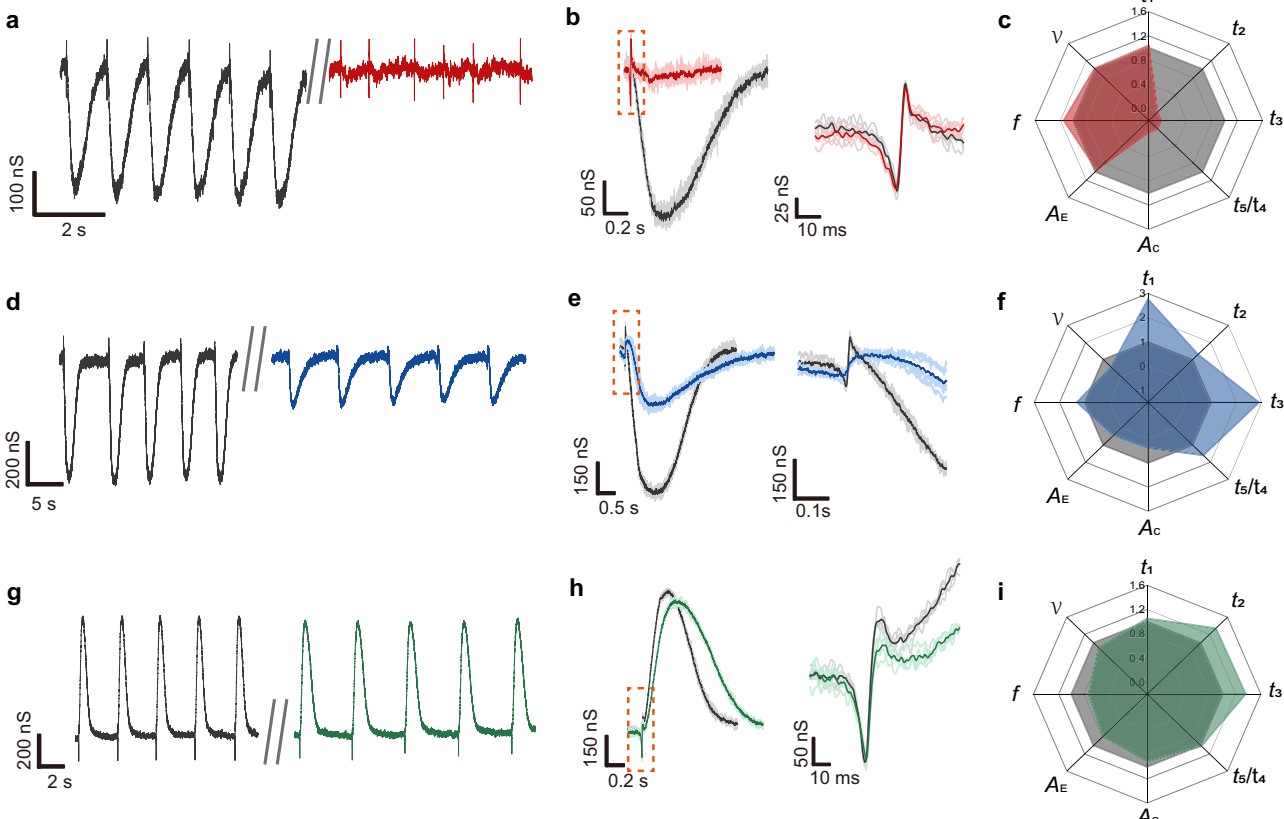

**Fig. 4 | Drug tests. a** Representative electrical recordings from a CMT before (black) and after (red) adding 10 μM blebbistatin. **b** (Left) Superimposed signals before and after adding blebbistatin. (Right) Zoom-in action potential signals from the dash box in left panel. **c** Radar map of the extracted parameters before (gray) and after (red) blebbistatin treatment. The parameters before the treatment were set as 1. **d** Representative electrical recordings from a CMT before (black) and after (blue) adding 1 μM Quinidine. **e** (Left) Superimposed signals before and after adding quinidine. (Right) Zoom-in action potential signals from the dash box in left panel. **f** Radar map of the extracted parameters before (gray) and after (blue) quinidine treatment. **g** Representative electrical recordings from a CMT before (black) and after (green) adding 1 μM doxorubicin. **h** (Left) Superimposed signals before and after adding doxorubicin. (Right) zoom-in action potential signals from the dash box in left panel. **i** Radar map of the extracted parameters before (gray) and after (green) doxorubicin treatment.

correlation, which was not possible with previous sensor technologies[21–29,31]. Meanwhile, the increasing rate of $Ca^{2+}$ uptake by the SR leads to a shortening relaxation phase in maturing cells[49], which was clearly reflected by the shortening $t_5$ (also contributing to the overall decrease in the entire contractile duration $t_2$). This unique capability of tracking high-resolution contractile details may be employed in studying cardiac aging, which can be subtly indicative of a prolonged relaxation in the contractile process without altering the force amplitude[11,12]. All these features were consistent with improving $Ca^{2+}$ handling and EC coupling during maturation. While post-dissection imaging revealed morphological evolution at sporadic stages to confirm the maturing process (Fig. 3f, g), the innervated mesh system enabled real-time tracking of the live CMT across the entire development and provided enriched physiological parameters beyond previous platforms.

## Drug tests

One of the main purposes of developing cardiac tissue models is to study drug effects because the heart is very susceptible to side effects from drugs[2]. Cardiotoxicity accounts for one-third adverse drug reactions and many are associated with ion-channel disturbance and result in arrhythmia[50]. The real-time tracking of the live CMT with enriched physiological parameters provides unique opportunity for the efficient detecting and differentiating drug effects. Drugs that work on different cell mechanisms were used to demonstrate the potential. Dimethyl sulfoxide (DMSO), the common carrier of drugs,

showed negligible effect on tissue state (Fig. S19), establishing the baseline for following tests. First, the innervated CMT was treated with Blebbistatin (10 μM), an inhibitor of the ATPase activity associated with the actin-myosin complex, to suppress contraction[51]. Electrical recordings showed a prominent suppression of the mechanical signals after the treatment (Figs. 4a, S20). Analysis of the action-potential signals showed negligible change in the amplitude $A_E$ and duration $t_1$ (Fig. 4b, c), suggesting a non-ionotropic mechanism consistent with the drug effect. Verapamil, another inhibitor of cell contraction, relies on the blockage of L-type $Ca^{2+}$ channels[52], which in turn also reduces the $Na^+$-channel activity[53]. As a result, the suppression of the mechanical contraction was coupled with weakened action potentials, which were reliably captured by electrical recordings (Fig. S21). These results show that the enriched physiological parameters provided by the innervated mesh system can discern mechanistic differences in close drug effects. The mesh-tissue intimacy also enabled the tracking of the evolution in drug effects. In one case, the mesh system provided a detailed comparison of the tissue state before and after recovery (e.g., washing out Blebbistatin) (Fig. S22). In another case, the mesh system revealed the dose-dependent drug effect of Verapamil, showing negative inotropic and chronotropic responses[31] with the increase of dosages (Fig. S23).

Conversely, we also constructed an electrically dysfunctional model by applying quinidine (1 μM), a $Na^+$-channel blocker[54], to the innervated CMT and recorded the signals (Fig. 4d). Examination of the overlayed signals revealed a decrease in action-potential amplitude $A_E$

but a prolongation in its duration $t_1$ (Fig. 4e, f). The trend was consistent with the drug mechanism in reducing and slowing $Na^+$ influx, which is the key component contributing to the recorded action-potential signal. The weakened $Na^+$-channel activity also affected the efficiency in $Ca^{2+}$ channels[55], leading to a decreased amplitude $A_C$ (Fig. 4d) and a prolongation $t_2$ in contraction (Fig. 4e, f). Dose-dependent studies enabled by the mesh further revealed that a dosage beyond 10 μM led to the full suppression of action potential and subsequent suppression of contraction (Fig. S24).

The above drug tests demonstrated that the innervated mesh system can track details to differentiate drug mechanisms that otherwise yield similar contractile impairment. The fine tracking may also help to capture subtle functional deviation. We therefore used the innervated sensing system to detect potential cardiotoxicity from doxorubicin, an effective anticancer agent commonly used for chemotherapy[56]. Although under normal dosage (e.g., <1 μM) the tissue maintained the basic electrical and mechanical functions (Fig. 4g, S25), a close examination of the recorded signals revealed changes in the EC dynamics (Fig. 4h). Specifically, although the decrease in the contractile amplitude $A_C$ was small (e.g., ~5%), a prominent prolongation in the contractile duration $t_2$ (e.g., >20%) was observed. The prolongation indicated the slowing in $Ca^{2+}$-handling dynamics, which was further suggested by the increased (~20%) EC delay $t_3$. However, unlike previous cases where the contractile prolongation mainly resided in the relaxation phase $t_5$ (Fig. 3c, 4e), here the contraction phase $t_4$ contributed proportionally (Fig. 4h, i). Meanwhile, the action potential only showed a slight decrease in the amplitude $A_E$, suggesting that an immediate ionotropic mechanism was not the direct cause. Although the mechanism is not fully known, the above observations are consistent with some existing proposed mechanisms, in which the doxorubicin produces a toxic metabolite that inhibits the sodium-calcium exchanger; this will further lead to a slowdown in the $Ca^{2+}$ uptake by sarcoplasmic reticulum and also a decrease in the L-type calcium channel activity[57], and hence, the prolongation in both relaxation and contraction phases. The innervated mesh not only readily captured the contractile deviation for identifying possible cardiotoxicity from doxorubicin, but also provided details for examining proposed molecular mechanisms.

## Disease modeling

We continued to demonstrate the potential of the innervated CMT for disease modeling. Myocardial infarction, which is usually caused by the occlusion of blood flow or deficient oxygen supply in local regions to result in cell malfunction or death, is the most common cause of heart failure[58]. The border zone between the infarcted and distal healthy tissue can undergo significant remodeling in the electrical and mechanical properties[59], which can eventually lead to heart failure. Tracking the EC-dynamics evolution in the border zone is crucial for the mechanistic understanding and eventual development of effective intervention and treatment[60]. While CMTs provide a convenient modeling platform, the typical use of an enclosed space to control the oxygen content creates a huddle for the real-time probing of the modeling result.

The innervated CMT is free of the huddle and can be readily placed in an enclosed microfluidic system capable of oxygen-flow control for the modeling study (Fig. 5a–c, S26). The CMT was subject to a hypoxic environment (10% oxygen) for 10 h before returning to a normoxia (21% oxygen) (Fig. S27) to emulate the conditions of ischemia and reperfusion, respectively[61]. The electrical recordings showed the development of tachycardia featuring a much faster (e.g., ~65% enhancement) beating rate after the hypoxic stress for 2 h (Fig. 5d–i, ii), which was attributed to the enhanced $Na^+$ current during hypoxia to result in $Ca^{2+}$ overload[62]. Within the next 4 h, the beating rate gradually reduced to value below the initial level (Fig. S28). Arrhythmia that features irregular beating was observed after 8-h hypoxic treatment

(Fig. 5d–iii), which was a characteristic symptom of ischemia[63]. Reestablishing a normoxic environment led to the gradual recovery of the beating rate (at 12 h, Fig. 5d–iv) close to the initial level (at 0 h). However, the recovery in the electrical conduction velocity, suggested by the temporal delay between the spatially distributed sensors (right panels, Fig. 5d), was slower after normoxia (Fig. S29). This suggested a decrease in the gap-junction Connexin-43 expression and an increase in tissue impedance consistent with previous studies[64]. The mechanical response recorded by channel-3 (blue curve) was completely suppressed after 8-h hypoxic treatment and did not recover by reperfusion (at 12 h), suggesting the irreversible loss of cell activity in the local region. As the locations of the corresponding sensors could be determined by later-stage imaging (Fig. S30), region-specific physiological activities could be captured to identify the ischemia-reperfusion injury[65]. Single-modal measurement such as electrophysiological recording alone is incapable of capturing such spatially differentiated EC decoupling effect for detailed tracking.

Analysis in the other parameters revealed additional information. A consistent trend of decreasing amplitude ($A_E$) and increasing duration ($t_1$) in the action potentials was observed after 6 h of hypoxia (Fig. 5e–i, ii), which was consistent with results from in vivo models of isolated guinea-pig ventricular myocytes and rabbit atrioventricular node. The result was also consistent with the mechanism of hypoxia-induced inactivation of the fast $Na^+$ channels[66], which can result in a decrease in both the amplitude and speed of the rapid depolarization phase in the action potential.

The mechanical contraction showed a slight decrease in the amplitude ($A_C$) but more prominent prolongation ($t_2$) that was mainly contributed from the contractile phase ($t_5$) after 6 h of hypoxia (Fig. 5e–iii–v; Fig. S31). Examination of the EC delay $t_3$ revealed a similar prolongation (Fig. 5e–vi; Fig. S32), suggesting a slowed triggering of the CICR process. The prolongations did not fully recover after the reperfusion (at 12 h). These features were expected from slowed $Ca^{2+}$-handling dynamics associated with the slowed upstroke velocity in the action potential (i.e., $t_1$ prolongation)[67]. Together, the lack of a full recovery in both the electrical and mechanical responses after the re-establishment of normal oxygen levels suggested that the cardiac tissue suffered irreversible damage during the process. The innervated mesh provided a window to observe the temporal and regional susceptibility of the cardiac tissue in response to hypoxic conditions. Capturing the differential details in response was essential for comprehending the complexities of cardiac function and developing targeted therapeutic strategies for ischemia-reperfusion injury.

## Discussion

In conclusion, our study has demonstrated a mesh sensing system with converged multifunctionality to track the detailed EC dynamics in cardiac tissues beyond previous platforms. The convergence of the multifunctional sensing in one device circumvents the challenges in signal synchronization, device addressing, and integration heterogeneity involved in the typical strategy of combining multiple sensor types. These budget reductions are very important for mesh integration and tissue interfacing, in which minimized invasiveness is always preferred. The mesh platform provides tissue-level softness and cell-level feature size to enable intimate interfacing with cardiac tissue. Together, the innervated sensing platform enables stable tracking of the enriched EC dynamics across the tissue and throughout the developmental process. These performance advances in the system enable comprehensive assessment for tracking cardiac tissue maturation, differentiating drug effects, and modeling disease as demonstrated in the study, providing enriched data sets far beyond what was possible with previous single-modal sensor technologies[21–31]. The sensing platform holds the promise to provide more accurate quantification of the functional, developmental, and pathophysiological states in cardiac tissues, creating an instrumental tool for

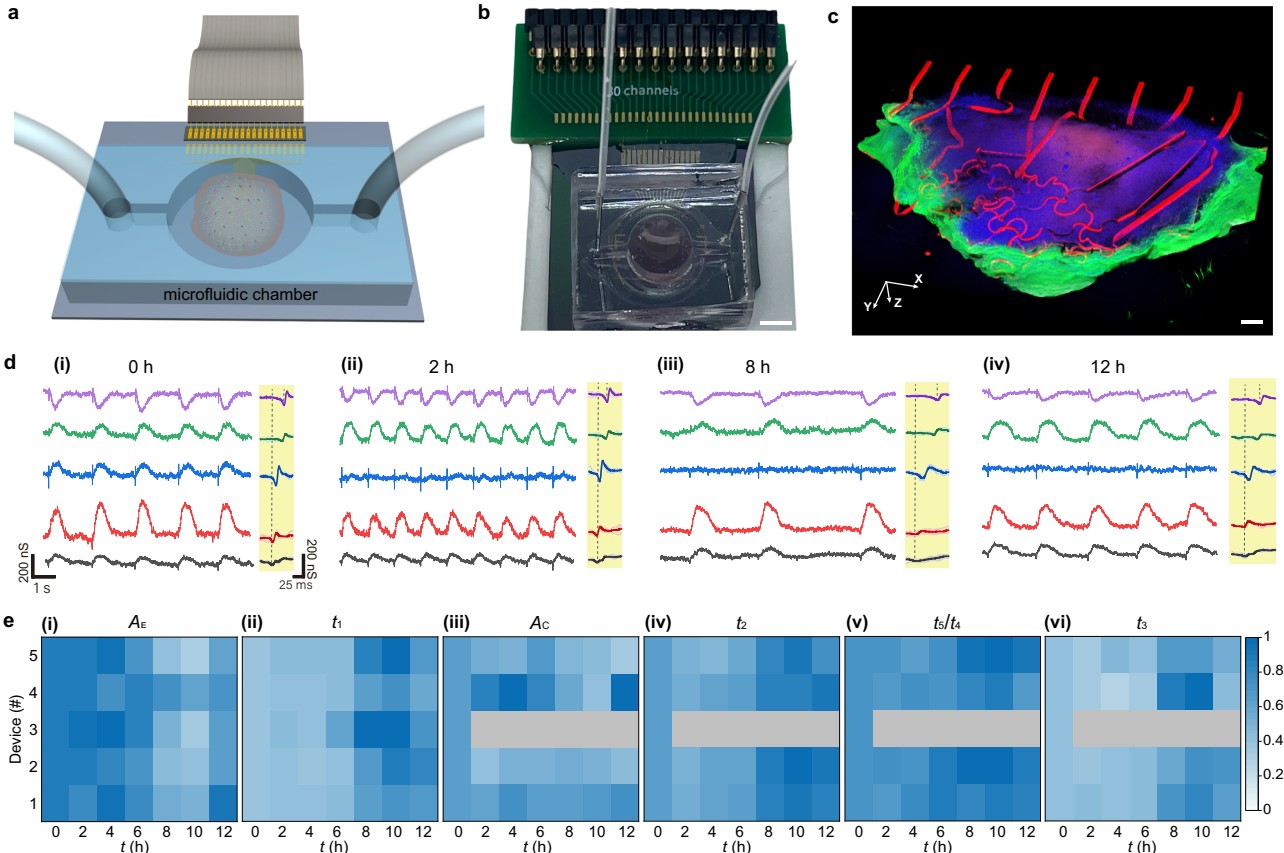

**Fig. 5 | Disease modeling. a** Schematic of a mesh-innervated CMT enclosed in a microfluidic system for disease modeling. **b** Optical image of the actual setup. Scale bar, 0.5 cm. **c** Reconstructed 3D fluorescence image of the (immunostained) mesh-innervated CMT (back-side). The red, green, blue, and purple colors correspond to mesh, F-actin, DAPI, and cTnT, respectively. Scale bar, 100 μm. **d** Electrical recordings of the CMT at different stages during hypoxia and normaxia. The right panel (yellow background) in each stage are the zoom-in superimposed action potentials. The recordings correspond (bottom to top) to graphene transistor sensors #1–5. **e** Evolution of the extracted parameters $A_E$, $t_1$, $A_C$, $t_2$, $t_5/t_4$, and $t_3$ as defined in Fig. 3a. All values were normalized to the largest value in each data group. Gray patterns represent the disappearance of the signal.

improving cardiac tissue engineering and studies. Previous studies have shown that graphene can maintain functional stability[68,69] and the mesh electronic scaffold can establish long-term interfacing intimacy[28,70–72] in in vivo tissue environments. These properties suggest that the graphene-integrated mesh platform, combined with efficient delivery and addressing strategy[73], can also be used in in vivo cardiac and neural systems for chronic recordings and studies. The platform also provides a unique tool to converge cell mechanics and electrophysiology studies—two very important but traditionally less overlapped fields.

## Methods

### Graphene synthesis

For monolayer-graphene synthesis, we used single-crystal Cu foil obtained by annealing polycrystalline Cu foil as the growth substrate. Graphene was then synthesized on the Cu foil in a low-pressure chemical vapor deposition (LPCVD) system using a modified two-step growth recipe[38] (Fig. S3). In the first step, the temperature was ramped from 25 °C to 1050 °C within 30 min. Then, 40 sccm of methane ($CH_4$) and 0.5 sccm of hydrogen ($H_2$) were introduced to the LPCVD system for the graphene synthesis. This resulted in the growth of multilayer graphene on the front side of the Cu foil. The system was maintained at 1050 °C for 40 min during this step. In the second step, the ratio of $CH_4$ to $H_2$ was changed to 1.5:40 (sccm), and the temperature was maintained at 1050 °C for 120 min. After this, the temperature gradually cooled down to 25 °C in 40 min. Adlayer-free, single-crystal graphene was synthesized on the backside of the Cu foil during this second step.

After the synthesis of graphene on Cu foil, a layer of PMMA 950 A4 (Kayaku Adv. Mater., Inc.) was spin-coated on the backside of the Cu foil and baked for 5 min at 80 °C. The frontside graphene was etched by oxygen plasma (50 W, 2 min, 50 sccm $O_2$). The sample was then floated on the Cu etchant (Transene, Inc.) for 30 min to remove the Cu foil. The released PMMA/graphene film was rinsed with deionized (DI) water and ready for the subsequent transfer process.

### Fabrication of the integrated mesh system

Detailed procedure is illustrated in Figs. S1, S2. The key steps include: (1) a patterned sacrificial layer (120-nm Ge) was deposited on the substrate ($Si/SiO_2$ or glass wafer). (2) A bottom SU-8 layer (2000.5; Kayaku Adv. Mater., Inc.) was spin-coated on the substrate and hard baked (For fluorescence imaging, 0.0008 wt% of Rhodamine 6 G power (Sigma-Aldrich) was mixed into the SU-8 solution). (3) The PMMA/Graphene film was then transferred onto the SU-8 layer, with the PMMA removed in acetone. (4) The graphene array was defined by lithographically patterning a protective resist array (PMGI SF6/S1805; Kayaku Adv. Mater., Inc.), oxygen-plasma etching (50 W, $O_2$ 50 sccm for 1 min) of unprotected graphene, and removal of the resist array. (5) A similar step was used to define a protective resist layer for defining the mesh pattern, etch the unprotected SU-8 film by reactive ion etching (RIE), and removal of the protective resist layer. (6) Standard photolithography and metal deposition were used to define the metal contacts and interconnects (Cr/Pd/Au/Pd, 3/15/40/5 nm). The contacts/interconnects were further passivated by a layer of $Si_3N_4$ (~70 nm). (7) The fabricated mesh system was released from the

substrate by etching the Ge layer in 1% $H_2O_2$ solution for 30 min and rinsed in DI water before cell culture.

## Raman spectrum

The Raman spectrum of the fabricated graphene devices was performed using a DXR Raman microscope (Thermo Fisher Scientific) with 600-nm excitation laser. The spectrum was recorded with an acquisition time of 40 s at a power of 4 mW.

## Electrical measurements

The water-gate effect and mechanical response in the graphene transistors were conducted in DPBS (Gibco) by using a semiconductor analyzer. The gate voltage was applied to a gold wire immersed in the DPBS solution. In the water-gate characterization, the conductance in the devices was measured under a fixed DC bias (10 mV) with a gate voltage (sweeping from 0 to 0.3 V) applied to the gold-wire gate electrode. The piezoresistive responses in the devices were performed by bending the substrate. Details can be found in Fig. S7. The in vitro electrical recordings from the cardiac tissues were performed with a DC bias (5–10 mV) applied to the drain. The drain current was amplified with a home-built amplifier system, and the output was acquired at 20 kHz by an A/D converter (Digidata 1440 A, Molecular Devices) interfaced with computerized software (pClamp 10.7, Molecular Devices). Data analysis was carried out using OriginPro (version 2022, OriginLab).

## Cell culture

hESCs (WAe009-A, H9, WiCell, NIH registration no. 0062, female. The cell line is commercially available and has been tested or authenticated by the manufacturers by standard techniques including morphology check, isoenzyme analysis, and mycoplasma detection as shown on their webpages: https://www.wicell.org/home/stem-cells/catalog-of-stem-cell-lines/wa09.cmsx) were used for the differentiation of cardiomyocytes by the following methods (Fig. S9)[40]. hESCs were maintained in the 60-mm tissue culture dishes coated with Matrigel (10 μg/ml, Corning) in DMEM-F12 (Gibco) using Essential 8 medium (Gibco) and subpassaged every 3 to 4 days. Cardiomyocyte differentiation from hESCs was conducted following the Wnt signaling modulation protocol[24,40]. During differentiation, hESCs were seeded in a 12-well plate for 2 to 3 days until 100% confluency. On day 0 of differentiation, the medium was changed to RPMI 1640 medium (Gibco) and 1% B27-insulin (Gibco) and 10 μM CHIR99021 (Tocris Bioscience). After 24 h (day 1), the medium was replaced to RPMI 1640 plus 1% B27-insulin. On day 3, the medium was changed to RPMI 1640 plus 1% B27-insulin, supplemented with 5 μM IWR-1-endo (Cayman Chemical). On days 5 and 7, the medium was changed to RPMI 1640 plus 1% B27-insulin, and RPMI 1640 plus 1% B27 (Gibco), respectively. The cells were maintained with RPMI 1640 plus 1% B27 every other day. The contraction of cells was usually observed on day 8. During days 10 to 12, the cardiomyocytes were ready for transfer and tissue culture.

For planar tissue culture (for device control test), the cells were rinsed with 1× DPBS to remove $Ca^{2+}$ and inhibit contraction. Cells were dissociated by treating 0.5 ml of 0.5 mM trypsin-EDTA (Ethlyenediaminetetraacetic acid, Gibco) per well for 4–5 min in a 37 °C incubator. The trypsin-EDTA was then aspirated, and cells were dissociated by gently pipetting with 2 ml RPMI 1640 plus 1% B27 using a 1-ml pipette tip. The cells were transferred to a 15-ml conical tube and centrifuged at 250 g for 3 min and then resuspended with 2 ml RPMI 1640 plus 1% B27 with 10 μM Y-27632 ROCK inhibitor (Tocris Bioscience). The substrate having the unreleased mesh was sterilized with 70% ethanol solution (1 h) and ultraviolet (1 h) and coated with Matrigel (20 μg/ml) in RPMI 1640 for 1 h at 37 °C. Cells were seeded on the device substrate at the density of $3–5 \times 10^5/cm^2$. The cells were maintained using RPMI 1640 plus 1% B27 with daily change. Electrical recordings were typically performed starting from day 7 after the cell seeding.

The innervation of the integrated mesh system into the CMT involved following steps: (1) Releasing the mesh system from substrate by etching the Ge sacrificial layer in 1% $H_2O_2$ solution for 30 min. (2) The released mesh was rinsed by DI water and sterilized by 70% ethanol for 1 h. (3) The mesh was washed with DPBS three times followed by the overnight treatment of Poly-D-lysine (0.1 mg/ml, Gibco) in incubator. (4) The Poly-D-lysine solution was aspirated, and the mesh was embedded in 100 μL Matrigel (10 mg/ml and incubated for 30 min to solidify), which also provided the initial fixation of the mesh on the substrate. (5) The resuspended cells in RPMI 1640 plus 1% B27 with 10 μM Y-27632 ROCK inhibitor were transferred onto the cured Matrigel with a density ($3–4 \times 10^6$ cells/cm²) and incubated for 24 h. Then the cells were maintained using RPMI 1640 plus 1% B27 by with the daily change of the medium. The electrical recordings typically started at day 5 of seeding (day 15–17 of the differentiation) and continually for more than 30 days.

## Calcium imaging

The mesh-innervated CMT in the chamber was rinsed with 1 ml of Hepes-buffered Tyrode's solution (Thermo Fisher Scientific) and stained with Fluo-8 AM (5 mM, AAT Bioquest). The CMT was then incubated for 1 h at 37 °C. Subsequently, the system was rinsed with Tyrode's solution three times. The $Ca^{2+}$ transient signal was recorded with a confocal microscope (Nikon A1R). Electrical recordings were performed simultaneously. The recorded signals were analyzed by ImageJ.

## CMT cryosection, immunostaining and imaging

CMTs at different culture days were rinsed with DPBS three times and fixed with 4% paraformaldehyde (PFA)/DPBS at 4 °C overnight. The CMTs were then transferred to 30% sucrose/DPBS solution for 24–48 h. The samples were embedded in optimal cutting temperature compound (Fisher Scientific) and cryosectioned into 30 μm-thick slides. For immunostaining, the CMT slices were treated with washing solution three times (0.2% Triton, Thermo Fisher Scientific) for 15 min each time. Slices were then washed in blocking solution (10% normal goat serum, Thermo Fisher Scientific) for 1–2 h in a humidified chamber at room temperature and subsequently replaced with primary antibody (Cardiac Troponin T Monoclonal Antibody Ctnt (clone number 13–11), diluted in the blocking solution with a ratio of 1:250, Invitrogen, Catalog # MA5-12960, LOT 2504985) diluted in blocking solution at 4 °C overnight. The secondary antibodies (Alexa Fluor™ 647, Goat anti-Mouse IgG (H + L) Cross-Adsorbed Secondary Antibody, diluted in the blocking solution with a ratio of 1:500, Invitrogen, Catalog # A-21235, LOT 1987304.), 4′,6-diamidino-2-phenylindole (DAPI, diluted in the blocking solution with a ratio of 1:1000, Invitrogen), and Phalloidin (Alexa Fluor™ 488, diluted in the blocking solution with a ratio of 1:50, Invitrogen) were co-stained for 1–2 h at room temperature. The slices were washed with DPBS three times before imaging. All samples were imaged by Nikon A1R confocal microscopy. Laser wavelengths of 405 nm, 488 nm, 561 nm, and 640 nm were applied to excite components labeled with DAPI, Phalloidin, Rhodamine 6 G, and cTNT, respectively. Images and fluorescence intensities were analyzed by NIS-Elements version 5.3.

## Whole-mount CMT immunostaining

The CMT was fixed with 4% PFA/DPBS solution at 4 °C overnight and washed with DPBS three times. The whole CMT was treated with 1 ml washing solution and 1 ml blocking solution with primary antibody (diluted in the blocking solution with a ratio of 1:250) at 4 °C overnight, respectively. The CMT was washed in blocking solution two times for 2 h and subsequently incubated with secondary antibody (diluted in blocking solution with a ratio of 1:500), DAPI (diluted in the blocking solution with a ratio of 1:1000), and Phalloidin (diluted in the blocking solution with a ratio of 1:50) overnight at 4 °C. Then the CMT was

washed in DPBS two times for 0.5 h and mounted with Fluoromount-G™ Mounting Medium (Invitrogen). The imaging of the whole CMT was performed by Nikon A1R confocal microscopy with a step of 5 μm and the 3D view of fluorescence images of CMT was reconstructed by NIS-Elements version 5.3.

## Drug assays

Drug assays were performed when the CMTs showed stable electrical and mechanical signals after culture for several days. Before drug testing, the medium was changed and the drug assays were conducted after 4 h of recovery. Drugs were dissolved in dimethyl sulfoxide (DMSO, Thermo Fisher Scientific) with a final concentration of less than 1%. Cumulative dosing of drugs was performed in the CMT until the recording signals presented stability. Drugs were purchased from Cayman Chemical.

## Integration with microfluidic system

The assembly of the microfluidic chamber and environment control system are illustrated in Figs. S26, S27 in detail. Briefly, the microfluidic chamber template/mold was laser cut from a poly(methyl methacrylate) plate (radius: 4.5 mm, height: 7 mm, volume: ~445 μL). The microfluidic chamber was fabricated by casting polydimethylsiloxane (PDMS, base: cure agent=10:1, Sylgard 184), and the released PDMS chamber was attached to the substrate with a thin layer of PDMS glue. The template/mold for the microfluidic chamber cap with micropillar arrays of different radii (0.5, 0.4, and 0.3 mm) was also laser cut from a poly(methyl methacrylate) plate and PDMS was casted to prepare the cap. The micropillar arrays were designed to eliminate the bubbles in the chamber[74]. The cap and chamber were bonded with double-coated spacer tape (ARseal 94119, Adhesives Research). Fluid was delivered to the channel using polyethylene tubes (BTPE-50, Instech), with the speed programmably controlled by a syringe pump (Harvard Apparatus, Standard Infuse/Withdraw Pump 11 Pico Plus Elite).

Hypoxic medium was prepared by bubbling ultrapure nitrogen and oxygen (9:1, Airgas) saturated with water to the culture medium (RPMI-1640 with HEPES (Gibco), pulsing 1% B27). The oxygen concentration was calibrated by an oxygen sensor (PCE Instruments). After 10-h hypoxia, the cell medium was replaced by normoxic medium. During the medium switching process, the flow rate was increased from 250 μL/h to 250 μL/min for ~2 min to flush the channel with required medium.

## Statistics and reproducibility

Microsoft Excel (version 2021) was used to analyze all data in this work. No statistical method was used to predetermine the sample size. No data were excluded from the analyses. The experiments were not randomized. The investigators were blinded to group allocation during data collection and analysis.

## Reporting summary

Further information on research design is available in the Nature Portfolio Reporting Summary linked to this article.

# Data availability

The data supporting the findings of this study are available within the article and its supplementary files. Any additional requests for information can be directed to, and will be fulfilled by, the corresponding author. Source data are provided in this paper.

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

## Acknowledgements

J.Y. acknowledges support from the Army Research Office W911NF2210027 and National Science Foundation (NSF) CBET-1844904. J.Y. also acknowledges supports from National Institutes of Health (NIH) 5R21 EB030216, NSF ECCS-1917630, and NSF DMR-2027102. Z.W. and J.K. acknowledges support from the Semiconductor Research Corporation Center 7 in JUMP 2.0 (award no. 145105-21913). Y.S. acknowledges support from NSF CMMI 1846866 and NIH R01DK129990. X.L. acknowledges support from a Link Foundation Energy Fellowship. Part of the device fabrication work was conducted in the clean room of the Center for Hierarchical Manufacturing (CHM), an NSF Nanoscale Science and Engineering Center (NSEC) located at the University of Massachusetts, Amherst.

## Author contributions

H.G. and J.Y. conceived the project. H.G. performed experiments in device fabrication, characterizations, cell culture, imaging, and in vitro electrical recordings. Z.W. and J.K. performed graphene synthesis. F.Y. and Y.S. contributed to cell supply and helped in cell culture. S.W. and X.W. assisted in device fabrication and cell culture. Q.Z. and X.L. assisted in microfluidic chamber fabrication. H.G. and J.Y. wrote the paper. All authors discussed the results and implications and commented on the manuscript.

## Competing interests

The authors declare no competing interests.
