## [Peer Review File · Nature Communications]

REVIEWER COMMENTS

Reviewer #2 (Remarks to the Author):

The study by Gao et al. introduces a mesh bioelectronic system embedded with multifunctional graphene nanoelectronic sensors for monitoring the excitation-contraction (EC) dynamics in cardiac microtissues—a key factor in cardiac disease modeling and developmental studies. This tissue-like system successfully overcomes the limitations of optical imaging and single-modal electronic sensing by combining both bioelectrical and biomechanical sensing to provide a comprehensive assessment of EC dynamics. Its tissue-level softness and cell-level feature size allow it to integrate seamlessly into microtissues, enabling stable, minimally invasive tracking throughout tissue development. The system's capabilities facilitate detailed monitoring of tissue maturation, drug effects, and disease modeling, offering a promising tool for enhancing accuracy in tissue engineering and research.

I believe this work is sufficiently different from the authors' previous publication (Sci Adv 8, eabn2485 (2022)) where the authors used suspended Si nanowires on a rigid 2D substrate to probe a thin layer of cultured cardiomyocytes. This current study leverages the flexibility and macroporosity of 3D mesh electronics to interface 3D microtissues that continuously expands and contracts. In addition, the use of graphene transistor offers greater mechanical robustness and chemical stability to interface with cardiac tissue. Therefore, I believe this work should be published in Nat Commun after some minor points are addressed.

1) How many independent recording channels can be incorporated into mesh electronics? Fig. 1h shows the data from 14 graphene transistors while Fig. 2d only shows 9 channels. Any particular reason for this difference in number of multiplexing?

2) How does the transconductance of graphene transistors compare with that of the authors' previously reported Si nanowire transistors?

3) Can the authors comment on the potential failure mode for chronic recordings and the possibility of applying their systems for in vivo cardiac and neuron recordings?

Signed off by G. Hong

Reviewer #3 (Remarks to the Author):

Gao et al. demonstrate a flexible mesh-system equipped with graphene transistors, allowing to sense electrical signals by the field effect, while mechanical strains are detected piezoelectrically. This presents a promising approach to concurrently record both types of signals on cardiac tissues, which the authors demonstrate through a comprehensive set of experiments. Notably, Gao et al. aim to disentangle the recorded signals into their electrically and mechanically caused proportion by their temporal difference. They showcase this approach by selectively inhibiting each type of signal using drugs, effectively showing how each contribution can be isolated and analyzed. In this context, the authors also discuss a possible cardiotoxicity from doxorubicin, which is inferred from the recorded signal features. It may be advisable to put this assessment into the context of results in literature, where the cardiotoxicity of doxorubicin and its underlying mechanisms have been studied (<https://doi.org/10.1016/j.biopha.2021.111708>). Finally, the authors should place a stronger emphasis on how this work differs from their previous work (<https://www.science.org/doi/10.1126/sciadv.abn2485>) and discuss the advantage of their graphene-based approach over nanowires.

Response to Reviewers

To make it clear, we have used *italic* fonts for the reviewers' comments, **black** fonts for our replies, and **blue** fonts for revisions.

Reviewer #1:

The authors of this study presented a 3D electrode array with a flexible graphene sheet to track multimodal excitation-contraction dynamics in cardiac microtissues. I believe that these results would be of interest to our subscribers, as this kind of tracking is necessary for drug screening. After the authors respond to a few minor questions, I recommend that this manuscript be accepted for publication.

We thank the reviewer for the time and effort devoted in the reviewing task and confirming the value of the work. Below please find our detailed addressing to each suggestion/question the reviewer raised.

In addition, the authors should consider the following comments:

1. The references section should include recent articles and review papers on 3D microelectrode arrays for in vitro cardiac tissue, as many such articles have been published recently.

We thank the reviewer for the suggestion. In the revised manuscript, we have added recent papers on 3D microelectrode arrays (MEAs) for *in-vitro* cardiac probing. Specifically, added Refs. 16-19 are representatives of 3D mushroom-shaped MEAs, nanopillar MEAs, nanocrown MEAs, and vertical transistor probes for cardiac recording. Ref. 17 (together with existing Ref. 15) is a review summarizing various 3D MEA probes, and Ref. 30 is a review summarizing integration methodologies for these probes.

16. Fendyur, A. & Spira, M. E. Toward on-chip, in-cell recordings from cultured cardiomyocytes by arrays of gold mushroom-shaped microelectrodes. *Front. Neuroeng.* **5**, 21 (2012).

17. Xie, C., Lin, Z., Hanson, L., Cui, Y. & Cui B. Intracellular recording of action potentials by nanopillar electroporation. *Nat. Nanotechnol.* **7**, 185-190 (2012).

18. Jahed, Z. et al. Nanocrown electrodes for parallel and robust intracellular recording of cardiomyocytes. *Nat. Commun.* **13**, 2253 (2022).

19. Gu, Y. et al. Three-dimensional transistor arrays for intra-and inter-cellular recording. *Nat Nanotechnol.* **17**, 292-300 (2022).

20. Abbott, J. et al. Optimizing Nanoelectrode Arrays for Scalable Intracellular Electrophysiology. *Acc. Chem. Res.* **51**, 3, 600–608 (2018).

30. Tang, X., He, Y. & Liu, J. Soft bioelectronics for cardiac interfaces. *Biophysics Rev.* **3**, 011301 (2022).

2. It is recommended that the authors provide a schematic timeline or protocol of the entire experiment, including cardiac differentiation and tracking of Excitation-Contraction Dynamics, in conjunction with Figure 2. This would help readers to better understand the study.

We thank the reviewer for the constructive suggestion. We have now added a schematic timeline for the entire experiment, including the differentiation of cardiomyocytes, integration of device with cardiac tissue, and tracking of excitation-contraction dynamics. We have also provided representative optical images during the differentiation of cardiomyocytes from hESCs. The schematic is added as a new Supplementary Fig. S9 (as shown below). Corresponding caption is also added for detailed specification.

Fig. S9. Timeline for mesh-CMT integration. **a**, Timeline for cell differentiation, integration of mesh electronics, and electrical recording. The differentiated cardiomyocytes were transferred onto the mesh during days 10 to 12. The mesh was gradually embedded into the CMT by tissue growth and folding process (Fig. 2b in main text). The electrical recording was started on days 15 to 17. **b**, Brightfield optical images showing increasing cell densities during the differentiation. Scale bar, 80 μm .

3. The authors need to provide more detailed information on how the synthesized monolayer graphene was transferred onto the SU-8 layer. While the fabrication steps are shown in S1, they are difficult to understand.

We thank the reviewer for the valuable suggestion. We have now added a schematic procedure and corresponding processing images in a new Supplementary Fig. 2 (see below) to detail the transfer.

Fig. S2. Transfer of graphene. **a**, Schematics of the transfer process. (i) Monolayer graphene grown on the backside of a Cu foil was spin-coated with a PMMA layer (~400 nm) and baked at 80°C for 5 min. The unprotected frontside graphene was etched away by oxygen plasma (50 W, 2 min, 50 sccm O₂). The sample was then floated on the Cu etchant (CE-100, Transene, Inc.) for 30 min to remove the Cu foil. (ii) The released PMMA/graphene film was rinsed with DI water for 10 times to remove Cu-etchant residue and stayed floating on DI water. (iii) The substrate coated with a SU-8 layer (hard-baked at 180°C for 30 min) was treated by oxygen plasma (50 W, 20 sccm O₂, 30 s) to make the surface hydrophilic. The substrate was then immersed in the DI water to pick up the PMMA/graphene film, and then baked at 100°C for 5 min to dry and improve the adhesion between graphene and SU-8. (iv) The PMMA film was removed by immersing the sample in acetone for 30 min. Subsequent fabrication processes as described in Fig. S1 were carried out for mesh fabrication. **b**, Actual steps of (i) removing the Cu foil by Cu etchant, (ii) cleaning the PMMA/graphene film by DI water, (iii) transferring the PMMA/graphene film onto the SU-8 substrate, and (iv) removing the PMMA film by acetone.

4. In Figures 2D-F, the mechanical motion tracking appears to have good reproducibility, while action potential tracking appears to be heterogeneous (Figure 2F), even when using the same CMT. What causes this heterogeneity in the data?

The recorded action potentials (APs) show common characteristic biphasic feature. The temporal difference between them (as expected) features the propagating conduction in the tissue. The difference in the details of the spikes (e.g., amplitude, duration, shape) is also expected, which can be contributed from several known factors:

(1) The differentiation procedure yields a mixture of different cardiomyocyte subtypes, including atrial, nodal, and ventricular-like cells. Different subtypes can exhibit different AP features, contributing to the shape difference observed in extracellular AP recordings (*STAR protocols* 3, 101560 (2022)).

(2) The three-dimensional cardiac microtissue can exhibit regional variances in structure and function, due to diffusion gradient in oxygen and nutrient. This variance can also contribute to variance in cell maturity that affects the AP duration. (*Sci Rep* 12, 17409 (2022); *Front. cardiovasc. med.* 6, 87 (2019)).

(3) Variation in cell-device proximity is known to affect AP amplitude.

We now have provided a brief explanation for the AP feature variation in the revised manuscript (page 6), with corresponding supporting references added:

“These sharp spikes showed a biphasic feature with the duration $\sim 27 \pm 6$ ms (**Fig. 2f**) and calibrated amplitude of ~ 70 -200 μ V (Fig. S10), consistent with the properties of the extracellular action potential in hESC-CMs.^{23, 26} The noticeable variation in the shape features of the recorded action potentials can be attributed from the mixed cardiomyocyte subtypes differentiated from hESCs and cell states affected by diffusion gradient in oxygen

and nutrient.^{41, 42} The time discrepancy between these action-potential signals (Fig. 2f) showed that the sensor array had the high temporal resolution to capture the propagating dynamics across different locations.”

5. *In Figure S13, the signal resolution and sensitivity (black and red) recorded by two graphene devices appear weaker than the Ca²⁺ signals from green fluorescence. Does this mean that the graphene device loses electrical signals while recording?*

We appreciate the reviewer’s careful observation. It is expected that geometric orientation in the mesh can have more prominent effect on the mechanical signal than the electrical AP signal—consider an extreme case when the contractile direction aligns with the mesh plane, and then the mechanical signal will be minimized whereas the AP signal shall be minimally affected. That is why we can see that the amplitude in the mechanical signals has a broader distribution than that in the AP signals (Fig. 2d). 1) The amplitudes of the mechanical signals recorded in Fig. S13 fell into the previous range. 2) Meanwhile, the recorded AP signals maintained a large amplitude. 3) No decay feature was observed in either the mechanical or electrical AP signals. These features suggest that the devices did not lose electrical signals during the recording. Of course, prolonged optical imaging at room temperature (~25 °C) can reduce cell activity in the tissue culture.

We have added the above points in the caption of Fig. S13 for clarification.

6. *The authors mention "improving sarcomere alignment" (Figures 3f, g), but it is unclear what this sentence means. Zoom-in images or immunofluorescence of sarcomere would be helpful.*

We thank the reviewer for the suggestion. We have included a zoom-in image to better illustrate the sarcomere ultrastructure (see inset in revised Fig. 3f as shown below). The sarcomere alignment is indicated by white arrows arrays.

7. The explanation of Figure S15 needs to be more detailed. What are the differences between 1, 2, and 3?

The main purpose of Fig. S15 (now Fig. S17) was to show that cell development can have regional differentiation (e.g., in timeline) and can be captured by the spatially distributed sensors. 1, 2, 3 represent signals recorded from three spatially distributed devices. In this case, signal 1 shows the early development of obvious mechanical contraction; signal 3 shows the beginning development of mechanical contraction; signal 2 shows the delayed development (e.g. no mechanical signal).

We have now added above clarification in the Fig. S17 caption.

8. In Figure S16, the authors mention that "the delay gradually decreases, indicating an improving electrical conduction velocity." What do the authors think that the decrease in delay is due to an increase in conduction velocity? More data is needed to support this claim.

We thank the reviewer for this question. A general understanding is that the action potentials in individual cells are initialized in a 'relay' format, passing along from one to another for the synchronized beating function. This leads to the electrical wave propagation/conduction. So the electrical conduction velocity will be $v = S/t$, where S is relative spatial distance between two cells and t the time delay between their initiations of

the action potentials. Since the relative spatial distance (S) between two sensors is largely fixed (indicated from a stable morphology in the tissue as shown below), then a reducing temporal delay (t) between the two acquired action-potential signals suggests an increasing conduction velocity.

In the revised Fig. S16 (now Fig. S18, see below), we have provided details to clarify and support above analysis.

Fig. S18 Temporal signal delay between different graphene devices. a-b, Optical images of the mesh-innervated CMT after 21 days and 41 days of development differentiation (days 10 and 30 of cell seeding). Scale bar, 200 μm . The red and yellow dashed lines delineate the boundary of CMT, showing minimal morphological change in the CMT during the continuous development. This also suggests that the relative spatial distance (S) between two embedded graphene devices does not change over time. c, Representative recordings from three devices at days 21, 28, 35, 42 of differentiation (days 10, 17, 24, 31 of cell seeding). The right panel shows zoom-in signals from the dash box in each recording. The (temporal) distance between each pair of dash lines defines the time delay (t) in action potentials recorded by the two graphene devices. Since the electrical conduction velocity can be calculated by $v = S/t$, the decreasing delay suggests an increasing electrical conduction velocity in the tissue.

Again, we sincerely thank the reviewer for the time, effort, and valuable suggestions to help us to improve the work. We believe we have addressed all for the improvement of our research quality for publication.

Reviewer #2 (Remarks to the Author):

The study by Gao et al. introduces a mesh bioelectronic system embedded with multifunctional graphene nanoelectronic sensors for monitoring the excitation-contraction (EC) dynamics in cardiac microtissues—a key factor in cardiac disease modeling and developmental studies. This tissue-like system successfully overcomes the limitations of optical imaging and single-modal electronic sensing by combining both bioelectrical and biomechanical sensing to provide a comprehensive assessment of EC dynamics. Its tissue-level softness and cell-level feature size allow it to integrate seamlessly into microtissues, enabling stable, minimally invasive tracking throughout tissue development. The system's capabilities facilitate detailed monitoring of tissue maturation, drug effects, and disease modeling, offering a promising tool for enhancing accuracy in tissue engineering and research.

I believe this work is sufficiently different from the authors' previous publication (Sci Adv 8, eabn2485 (2022)) where the authors used suspended Si nanowires on a rigid 2D substrate to probe a thin layer of cultured cardiomyocytes. This current study leverages the flexibility and macroporosity of 3D mesh electronics to interface 3D microtissues that continuously expands and contracts. In addition, the use of graphene transistor offers greater mechanical robustness and chemical stability to interface with cardiac tissue. Therefore, I believe this work should be published in Nat Commun after some minor points are addressed.

We thank the reviewer for taking time & effort for the review task, in particular, confirming the quality/value of the work by providing a succinct and savvy summary. We also thank the reviewer for raising valuable comments to help us to improve the scientific presentation. Below please find our detailed response to each of the comments.

1) How many independent recording channels can be incorporated into mesh electronics? Fig. 1h shows the data from 14 graphene transistors while Fig. 2d only shows 9 channels. Any particular reason for this difference in number of multiplexing?

We appreciate the reviewer's careful observation. In principle, we can integrate more transistors in the mesh (e.g., by reducing ribbon feature and pitch size). However, we have a limited number of (home-built) current amplifier channels (e.g., 12) for simultaneous recordings. Thus, integrating 14 transistors on the mesh is a reasonable design (by also taking device yield into account) without losing the purpose of validating system function. Our as-fabricated mesh system generally can achieve >90% device yield.

The actual recording yield (9/14 ~70%) reduced after the integration with cardiac microtissue can come from several known factors:

(1) The *Wnt* signaling protocol employed for cardiomyocyte differentiation from hPSCs is known to yield 80-98% functional cardiomyocytes (Nat Protoc 8, 162–175 (2013)). Without additional cell sorting during cell seeding on mesh, we anticipate that this yield will carry on to successful device-cell interfacing.

(2) Without an effective vascular system, the supply of nutrient and oxygen by diffusion in 3D microtissue is limited, leading to possible apoptotic/necrotic inner core (Sci Rep 12, 17409 (2022)).

In the revised manuscript, we have added specifications in the manuscript (page 6) to elucidate the above points: “The recording yield (e.g., 9/14 ~ 64%) was reasonable by taking consideration of yields in device fabrication (e.g., ~90-95%), functional cardiomyocytes (e.g., 80-98%) during differentiation,⁴⁰ and potential inner necrotic part in the CMT resulted from supply limit of nutrient and oxygen by diffusion.⁴¹”

2) How does the transconductance of graphene transistors compare with that of the authors' previously reported Si nanowire transistors?

We appreciate the reviewer's question. The maximal transconductance of graphene transistors (Fig. 1h and Fig. S4) is $\sim 2.2 \pm 0.4$ mS/V, which is around 500 times larger than the average transconductance of suspended Si nanowire $\sim 4.2 \pm 1.0$ μ S/V in our previous work. Taking the noise level into consideration, the estimated detection thresholds for graphene and Si nanowire transistors are ~ 30 - 60 μ V and ≥ 200 μ V (e.g., with a signal-to-noise ratio >3), respectively.

We have now added this comparison in the revised manuscript.

3) Can the authors comment on the potential failure mode for chronic recordings and the possibility of applying their systems for *in vivo* cardiac and neuron recordings?

We thank the reviewer for the question. The potential failure of chronic recordings may come from three foreseeable main factors: (1) the aging of tissue, (2) the degradation in graphene device quality, and (3) reduced intimacy in device-tissue interface. Item (1) is extrinsic to the recording itself, so we may focus on items 2&3 for discussion.

First, graphene has been studied to be stable in physiological/tissue environment due to its chemical stability. In our test window (e.g., chronic recording at day 42 after differentiation), stable electrical and mechanical responses were maintained. Other experimental study showed stable recording (only electrical signal) beyond 100 days. Therefore, (with the one-face protection by substrate and possible edge passivation) we expect that a graphene transistor in itself can maintain stability (e.g., beyond months) suitable for many chronic studies.

Second, existing *in vivo* studies on mesh electronics (*Nat Mater* **18**, 510–517 (2019); *Nat Methods* **13**, 875–882 (2016); *Nat Neurosci* **26**, 696–710 (2023)) have shown that the ribbon feature in the mesh can help maintain an intimate and stable device-cell interface beyond the time span of a year.

Putting together, we may project that graphene-integrated mesh electronics can attain similar stability to metal-integrated mesh electronics for *in vivo* recordings. The added benefit may include 1) the transparency in graphene can facilitate concurrent optical/optoelectronic/optogenetic functions; 2) the mechanical sensing (obviously it's useful for cardiac system) might be also useful in neural system (e.g., detection change of intracranial pressure).

In the revised manuscript, we have included a discussion on the potential application of graphene-integrated mesh electronics for *in vivo* cardiac and neural recordings (with above new references added): "Previous studies have shown that graphene can maintain functional stability^{68,69} and the mesh electronic scaffold can establish long-term interfacing intimacy^{28,70-72} in *in vivo* tissue environments. These properties suggest that the graphene-integrated mesh platform, combined with efficient delivery and addressing strategy,⁷³ can also be used in *in vivo* cardiac and neural systems for chronic recordings and studies."

Again, we would like to thank the reviewer for confirming the value in our work and all the constructive suggestions to help improve the presentation.

Reviewer #3 (Remarks to the Author):

Gao et al. demonstrate a flexible mesh-system equipped with graphene transistors, allowing to sense electrical signals by the field effect, while mechanical strains are detected piezoelectrically. This presents a promising approach to concurrently record both types of signals on cardiac tissues, which the authors demonstrate through a comprehensive set of experiments. Notably, Gao et al. aim to disentangle the recorded signals into their electrically and mechanically caused proportion by their temporal difference. They showcase this approach by selectively inhibiting each type of signal using drugs, effectively showing how each contribution can be isolated and analyzed.

We thank the reviewer for confirming the general interest in our work and raising constructive suggestions to help improve the scientific presentation. Below please find our detailed response to each of the suggestions.

In this context, the authors also discuss a possible cardiotoxicity from doxorubicin, which is inferred from the recorded signal features. It may be advisable to put this assessment into the context of results in literature, where the cardiotoxicity of doxorubicin and its underlying mechanisms have been studied (<https://doi.org/10.1016/j.biopha.2021.111708>).

We really thank the reviewer for pointing us to this valuable review, which provides information we previously did not know of. The review certainly covers multilevel (proposed) mechanisms regarding the cardiotoxicity from doxorubicin. We notice that in one mechanistic picture, it describes that the metabolite from doxorubicin leads to inhibition effect in the sodium-calcium exchanger – this slows down the Ca²⁺ uptake by the sarcoplasmic reticulum, as well as the L-type calcium channel activity. Consequently, prolongation in both the relaxation and contraction phases is expected. This is consistent with our experimental observation that we observed a proportional ~20% prolongation in both phases.

We have now added following description by referring to this added ref. (page 10): “... unlike previous cases where the contractile prolongation mainly resided in the relaxation phase t_5 (Figs. 3c, 4e), here the contraction phase t_4 contributed proportionally (Figs. 4h, i). Meanwhile, the action potential only showed a slight decrease in the amplitude A_E , suggesting that an immediate ionotropic mechanism was not the direct cause. Although the mechanism is not fully known,⁵⁷ above observations are consistent with some existing proposed mechanism, in which the doxorubicin produces toxic metabolite that inhibits the sodium-calcium exchanger; this will further lead to a slowdown in Ca²⁺ uptake by sarcoplasmic reticulum and also a decrease in the L-type calcium channel activity,⁵⁷ and hence, the prolongation in both relaxation and contraction phases. The innervated mesh not only readily captured the contractile deviation for identifying possible cardiotoxicity from doxorubicin, but also provided details for examining proposed molecular mechanisms.”

Finally, the authors should place a stronger emphasis on how this work differs from their previous work (<https://www.science.org/doi/10.1126/sciadv.abn2485>) and discuss the advantage of their graphene-based approach over nanowires.

We thank the reviewer for this very constructive suggestion to help readers to appreciate the difference and advance. We have summarized the advances as follows:

1. Previous nanowire work only demonstrated planar/rigid-substrate integration and the detections were limited to planar tissue culture. It is nontrivial (in fact difficult) to achieve 3D nanowire assembly on a freestanding mesh for 3D tissue interfacing/recording.

Over the long run, the intrinsic randomness in the bottom-up nanowires will inherently limit the scalability in the technology.

2. The graphene devices featured a much higher transconductance and uniformity ($\sim 2.2 \pm 0.4$ mS/V), compared to values in nanowire devices ($\sim 4.2 \pm 1.0$ μ S/V). These enhanced electronic properties can benefit signal detection (e.g., detection limit) and data representation/interpretation.

We have now highlighted above points by adding a new paragraph in the manuscript (page 4): “Recently, we developed a 2-in-1 sensor concept of converging electrical and mechanical sensing in one device, by exploiting the field effect and piezoresistive effect in a bottom-up semiconducting silicon nanowire.³⁴ The nanowire was specifically assembled into 3D suspended structure on a planar substrate to attain geometric freedom for enabling multifunctionality. Both electrical action potential and mechanical contraction were simultaneously detected in a planar cardiac tissue culture. However, the device integration in CMT has not yet been achieved, given that attaining scalable 3D nanowire assembly from the bottom up on a freestanding substrate can be nontrivial.”

We have also added other descriptions in the scalability and electronic properties of graphene (in comparison with nanowires) across the result section to highlight the long-term benefit of the integrated system.

In the end, we would like to thank the reviewer again for confirming the value in our work and the constructive advice to help us to improve the quality.

REVIEWERS' COMMENTS

Reviewer #2 (Remarks to the Author):

I support the acceptance of this paper.

Reviewer #3 (Remarks to the Author):

The authors have addressed all comments satisfactorily.